# Evaluation of the insecticide custody chain and its relationship with malaria burden in the Brazilian Amazon: a process and exploratory impact assessment (2019-2023)

**Marcela Lima Dourado**[1,2/+], **Rafaella Albuquerque e Silva**[2], **Márcia Caldas de Castro**[3], **Cássio Roberto Leonel Peterka**[2,4], **Carolina Ribas Kluge**[2], **Daniele Castro**[1], **José Bento Pereira Lima**[1]

[1]Fundação Oswaldo Cruz-Fiocruz, Instituto Oswaldo Cruz, Rio de Janeiro, RJ, Brasil
[2]Ministério da Saúde, Brasília, DF, Brasil
[3]Harvard University, Harvard TH Chan School of Public Health, Boston, MA, Estados Unidos da América
[4]Secretaria de Saúde do Estado do Amapá, Macapá, AP, Brasil

**BACKGROUND** Malaria remains a major public health concern in Brazil, with the Amazon region accounting for 99.9% of the country's cases. Indoor residual spraying (IRS) using Etofenprox 20% PM is a core vector control strategy. However, inefficiencies in the insecticide custody chain, including planning, storage, and distribution, may compromise intervention effectiveness.

**OBJECTIVES** This study aimed to evaluate the insecticide custody chain from a process perspective, identifying logistical bottlenecks, while also exploring potential impact by examining associations between insecticide allocation and malaria burden in Brazil and in three high-incidence municipalities in the state of Amazonas (Barcelos, Tefé, and São Gabriel da Cachoeira) from 2019 to 2023. The underlying hypothesis is that in a well-functioning system, insecticide distribution should be correlated with malaria risk determinants, such as epidemiological and environmental variables, rather than merely responding to reported case counts.

**METHODS** A mixed-methods approach was used. Quantitative analysis applied Pearson correlation, simple and multiple linear regressions (with time-lag), and ARIMA models to evaluate associations between insecticide volume and malaria cases, incorporating environmental and demographic covariates. A complementary qualitative assessment, based on a structured risk matrix, examined failures across four stages of the custody chain: planning, storage, application, and monitoring.

**FINDINGS** At the national level, insecticide volume was significantly associated with malaria cases ($\beta = 0.161$; $p = 0.038$; $R^2 = 0.74$) and deforestation ($\beta = 0.626$; $p = 0.034$). Time-series analysis revealed a reactive pattern, with insecticide allocation often lagging behind malaria incidence peaks. In the municipalities studied, models lacked statistical significance, but trends suggested weak local planning and disconnects from risk-based forecasting. The risk matrix revealed systemic weaknesses, including limited data interoperability, insufficient integration of environmental indicators, and poor federal coordination.

**MAIN CONCLUSIONS** The current custody chain functions reactively and lacks integration with surveillance and predictive environmental data, contradicting World Health Organization (WHO) recommendations. Structural reforms are urgently needed. These include: (i) risk-based planning incorporating environmental variables, (ii) interoperable information systems, (iii) improved surveillance of vector resistance, and (iv) intergovernmental agreements for equitable and efficient resource allocation. Findings highlight the need for strategic reorientation of IRS logistics toward anticipatory and data-driven planning. Strengthening the custody chain through intersectoral coordination and environmental intelligence is essential not only to improve operational efficiency but also to increase the cost-effectiveness and epidemiological impact of malaria control interventions in the Amazon. Lessons learned may inform broader efforts in other endemic regions aiming for malaria elimination.

Key words: malaria - supply chain - indoor residual spraying - insecticides - process evaluation - vector control - deforestation - public health surveillance - Amazon

Malaria remains one of the most persistent endemic diseases in the Brazilian Amazon, accounting for over 99% of the country's autochthonous cases.[1] Although Brazil has made significant progress in controlling the disease over recent decades, the persistence of active foci, particularly in remote areas with high socio-environmental vulnerability, poses a substantial challenge to the sustainability of these achievements. In response to this scenario, the Brazilian Ministry of Health launched the National Malaria Elimination Plan (PNEM) in 2022, setting a target to eliminate autochthonous transmission by 2035[1] [Supplementary data (Fig. 1)].

The national elimination strategy is structured into four sequential phases and organized around three priority pillars: (i) qualified epidemiological and entomological surveillance; (ii) intensified vector control measures; and (iii) institutional capacity strengthening. The strategy also incorporates risk stratification and territorial adaptation, in alignment with recommendations from the World Health Organization (WHO).[1]

+ Corresponding author: marcelalima88@gmail.com | ⊙ https://orcid.org/0009-0007-1193-0122

**Handling editor:** Adeilton Alves Brandão| ⊙ https://orcid.org/0000-0001-5877-607X

Among vector control tools, indoor residual spraying (IRS) using pyrethroid insecticides remains one of the principal strategies implemented by the National Malaria Control Program (PNCM)[2] [Supplementary data (Figs 2-3)]. However, the effectiveness of IRS depends heavily on an efficient supply chain, demand forecasting, and coordination across the multiple levels of governance within the Brazilian Unified Health System (SUS). The absence of predictive planning and the reactive nature of supply requests undermine the rational use of public resources and reduce the impact of interventions.[3,4]

According to PNCM protocols and WHO guidelines,[3] insecticide allocation should be guided by entomological risk, climatic seasonality, vector density, and epidemiological history to anticipate outbreaks and optimise IRS coverage. In practice, however, planning often remains reactive, responding mainly to increases in case numbers, which undermines the effectiveness of vector control measures. Beyond these operational parameters, factors such as deforestation have been highlighted in the scientific literature as important determinants of malaria transmission. While not included in official operational guidelines, we propose deforestation as a complementary predictive variable to strengthen risk-based planning within the custody chain framework.

Despite the recognition of IRS as a key strategy to interrupt malaria transmission, few studies have systematically evaluated the custody chain of insecticides, particularly within the Amazonian context. The national literature lacks integrated analyses that combine logistical, environmental, and epidemiological indicators to assess the responsiveness and operational effectiveness of this supply chain in meeting local vector control needs.

This study is based on the hypothesis that the current custody chain of Etofenprox 20% PM operates in a decentralized and weakly integrated manner with disease surveillance systems, resulting in structural deficiencies that impair its effectiveness and predictive capacity. It is further hypothesized that environmental factors, such as deforestation and climatic variation, are not adequately incorporated into IRS planning, resulting in reactive and inefficient responses. Therefore, this study aims to fill this gap by conducting an integrated analysis of the insecticide custody chain, grounded in empirical data and evidence-based insights.

Recent studies have shown that environmental factors, such as temperature, rainfall, and deforestation, significantly influence malaria transmission dynamics and should therefore inform the planning of vector control activities.[5,6] However, the logistical governance of the insecticide custody chain in Brazil remains fragmented and largely driven by decentralized requests, without systematic integration of these variables.

Furthermore, it should be noted that in this study, the term 'chain of custody' is used in a broad sense. It encompasses not only product integrity (*i.e.*, storage and transport conditions), but also the institutional, operational and logistical mechanisms that ensure the timely, efficient and traceable delivery of inputs from acquisition to application in the field. This definition aligns with WHO guidelines on public health supply chain management, broadening the traditional notion of custody to encompass federal coordination, data integration, and the governance of decision-making processes.[1]

In this context, the present study aims to evaluate the custody chain of Etofenprox 20% PM used in IRS from 2019 to 2023, focusing on three high-priority municipalities in the state of Amazonas, Barcelos, Tefé, and São Gabriel da Cachoeira [Supplementary data (Fig. 4)]. The analysis examines the relationship between insecticide dispensation, epidemiological trends, climatic and environmental variables, with the goal of proposing evidence-based strategies to improve vector control policy and support Brazil's malaria elimination targets.

## MATERIALS AND METHODS

This study is primarily designed as a process evaluation, focused on identifying operational bottlenecks within the custody chain of Etofenprox 20% PM insecticide used in malaria vector control. Additionally, it integrates elements of an exploratory impact assessment to investigate potential associations between insecticide allocation and malaria incidence [Supplementary data (Tables I-II)].

The central objective is to examine whether the planning and distribution of insecticide inputs are based on proactive strategies, anticipating environmental and epidemiological risk factors, or whether they reflect reactive responses to already elevated case counts. Accordingly, the underlying hypothesis posits that a well-functioning custody chain should exhibit a predictive and risk-based allocation pattern, with a meaningful correlation between input volumes and malaria transmission determinants, rather than a delayed response to outbreaks.

To achieve this, the study adopts a mixed-methods approach that combines process evaluation, assessing the operational structure, logistical integrity, and strategic planning of the insecticide distribution system, with exploratory impact analysis, through the application of regression modeling to examine the relationship between the volume of insecticide distributed and the malaria burden in selected municipalities of the Brazilian Amazon.

The research hypotheses that guided this investigation were as follows:

(H1) At the national level, the distribution of insecticides is significantly associated with the number of malaria cases, reflecting a reactive approach to logistical planning.

(H2) At the municipal level, however, this association tends to be weak or non-existent due to fragmented local management and an absence of predictive planning.

(H3) Environmental factors such as deforestation and temperature influence the malaria burden and should therefore be incorporated as predictive variables in vector control programming.

(H4) We thank the reviewer for this observation. We have softened the language throughout the manuscript by replacing the term *"effectiveness"* with *"operational efficiency/potential reach"*. Direct conclusions were removed, and we now explicitly use the term *"exploratory"* whenever potential impacts are mentioned, to align with the scope and limitations of the study.

This is an observational, ecological study employing a mixed-methods approach, which evaluated the custody chain of the residual insecticide Etofenprox 20% PM used in IRS as a strategy for malaria vector control. The analysis covered the period from 2019 to 2023 and included both national-level data and municipal-level case studies, focusing on three high-burden localities in the Brazilian Amazon: Barcelos, Tefé, and São Gabriel da Cachoeira, all located in the state of Amazonas [Supplementary data (Figs 4-6)].

For the quantitative analysis, the municipalities were selected based on their epidemiological relevance and the completeness of available data during the study period [Supplementary data (Figs 4-5)]. Years and localities with incomplete or inconsistent records in official systems were excluded from the comparative analysis. Specifically, data on insecticide dispensation for São Gabriel da Cachoeira in 2022 were excluded, as the recorded value was zero, an outlier that deviated from expected patterns, and there was no documented suspension of vector control activities for that period. Similarly, missing values for climatic and demographic variables were either imputed using the annual moving average or left missing when such imputation lacked statistical justification. This strategy aimed to ensure greater analytical robustness and avoid interpretive bias arising from artificial data insertion.

*Data sources* - Secondary data were obtained from official information systems of the Brazilian Ministry of Health and national research and monitoring institutions:

• Malaria cases: Malaria Epidemiological Surveillance Information System (Sivep-Malaria) [Supplementary data (Table III)];

• Insecticide dispensation: Strategic Health Supplies Information System (SIES/MS) and Amazonas State Health Surveillance Foundation (FVS-AM) [Supplementary data (Tables IV-V)];

• Population data: Brazilian Institute of Geography and Statistics (IBGE), intercensal estimates [Supplementary data (Table VI)];

• Environmental data (temperature and precipitation): National Institute of Meteorology (INMET) [Supplementary data (Table VII)];

• Deforestation data: TerraBrasilis platform, National Institute for Space Research (INPE) [Supplementary data (Table VIII)].

*Indicators and variables* - The main variables analyzed included one dependent variable, the number of reported malaria cases per year and per municipality, and several independent variables, namely: the total volume of insecticide dispensed (kg), average annual temperature (ºC), average annual precipitation (mm), total deforested area (km²), and estimated population by municipality [Supplementary data (Table VI)].

*Statistical analysis* - A descriptive analysis of the temporal data was conducted at both national and municipal levels. Subsequently, Pearson correlation analysis was applied to assess the relationships between malaria cases and the independent variables, with statistical significance set at $p < 0.05$ [Supplementary data (Figs 7-10)]. To investigate the associations between explanatory variables (volume of insecticide, temperature, precipitation, and deforestation) and the response variable (number of malaria cases), simple linear regression models were fitted [Supplementary data (Table IX)].

For the aggregated national data, linear regression models with time-lag incorporation (1-year lag) were also applied, in addition to univariate ARIMA models for time series analysis [Supplementary data (Fig. 11)].

All models were evaluated for statistical significance (p-values), adjusted coefficient of determination ($R^2$), 95% confidence intervals (95% CI), and residual consistency. The analyses were performed using Python 3.11 and the Statsmodels library version 0.13.5 (Table I).

The Pandas libraries were used for data structuring, Statsmodels 0.13.5 for linear regressions and pmdarima for time series modelling. Missing data in the climate variables were imputed using a 3-year moving average, provided the temporal variance was less than 10%. In cases of greater variability, the missing data were retained to avoid bias. The model residuals were examined for normality, autocorrelation and homoscedasticity.

*Complementary qualitative analysis* - In addition to the quantitative analysis, risk matrix techniques were employed to evaluate the custody chain across four strategic components: planning, storage, application, and monitoring. Risk classification was based on the frequency and impact of observed failures, using parameters defined by the Brazilian Ministry of Health and guidelines from the WHO [Supplementary data (Table X)].

*Process* - This refers to the main operational stages of the insecticide supply chain, from demand forecasting to the monitoring of outcomes. The processes analyzed included: planning and demand programming, product procurement and quality assurance, storage and transport, distribution and application, and monitoring and evaluation.

TABLE I

Classification into high, medium, or low was based on a qualitative assessment of historical frequency

| Level | Description |
| --- | --- |
| High | High likelihood of occurrence; failures observed or expected frequently. |
| Medium | Moderate probability; depends on specific conditions to occur. |
| Low | Low probability; effective controls already implemented. |

*Identified risks* - These refer to potential failures or threats that may compromise the performance of each stage in the process. Risks may be associated with operational, logistical, financial, human, or structural factors. Key risks identified included inaccurate demand forecasting and delays in procurement processes. For the classification of probability of occurrence, three levels were used: low, medium, and high disease burden. Examples include:

• Inaccurate demand estimation;
• Procurement delays;
• Inadequate storage infrastructure;
• Workforce shortages;
• Insufficient monitoring.

*Probability* - This refers to the likelihood of a given risk occurring. Classification into high, medium, or low was based on a qualitative assessment of historical frequency, process vulnerability, and the presence (or absence) of preventive controls (Table I).

*Impact* - This assesses the consequences if the identified risk materializes. Impact was measured in terms of public health outcomes, financial losses, logistical disruptions, or reduced effectiveness of vector control actions (Table II).

*Mitigation measures* - These refer to proposed preventive or corrective actions aimed at reducing the probability or impact of a given risk. Examples include: enhancements to information systems, streamlining procurement procedures, staff training, infrastructure investments, and the implementation of monitoring tools.

*Risk level calculation and classification* - The overall level of risk was determined by combining probability and impact using a 3x3 matrix (high, medium, low) (Table III).

Risk matrix of the insecticide custody chain in malaria control in Brazil. Presents logistical processes, identified risks, levels of probability and impact (high, medium, or low), and corresponding mitigation measures.

Thus, a risk with high probability and high impact is classified as a high risk, requiring priority action. Conversely, a risk with medium probability and medium impact is considered moderate risk, warranting attention but with lower urgency.[7]

The matrix highlights critical points in the supply chain, such as failures in predictive planning, lack of integration between information systems (Sivep-Malaria, SIES, and FVS), low traceability of supplies,

absence of systematic monitoring of vector resistance, and weak federal coordination. These factors increase the risk of stockouts, misallocation of resources, and reduced effective coverage of IRS, thereby jeopardizing the goals of the PNEM.[4,7]

## RESULTS

Between 2019 and 2023, Brazil recorded a 26% reduction in the number of reported malaria cases, decreasing from 157,454 to 116,147 cases. During the same period, approximately 609,115 kg of Etofenprox 20% PM insecticide were acquired and distributed for use in IRS, at an estimated cost of BRL 32.7 million. A 65% reduction in the total quantity of insecticide dispensed was observed over the period [Supplementary data (Tables IV-V)].[8,9]

Pearson correlation analysis showed a moderate association between the amount of insecticide distributed and the annual number of malaria cases ($r = 0.52$), along with a negative correlation between average annual temperature and insecticide volume ($r = -0.55$) [Supplementary data (Figs 7-10)].

Simple linear regression analysis demonstrated a statistically significant association between the amount of insecticide dispensed and the number of malaria cases ($\beta = 0.161$; $p = 0.038$; 95% CI: 0.015-0.308; adjusted $R^2 = 0.74$). Similarly, cumulative deforestation showed a positive correlation with malaria incidence ($\beta = 0.626$; $p = 0.034$; 95% CI: 0.090-1.163; adjusted $R^2 = 0.76$). Average precipitation showed a trend toward an inverse association, though not statistically significant ($\beta = 0.228$; $p = 0.057$). Average annual temperature did not show a significant association (Fig. 1).

Furthermore, it is worth noting that the ARIMA models fitted to the national time series demonstrated stationary behavior after first-order differencing, with independent residuals and an approximately normal distribution, indicating statistical adequacy. The analysis revealed that increases in insecticide dispensation frequently occurred after peaks in malaria incidence, suggesting a reactive pattern in logistical management. Additionally, in the municipalities analyzed, the models indicated that fluctuations in temperature and deforestation preceded changes in malaria cases by a lag of one to two months, reinforcing their potential as predictive variables [Supplementary data (Figs 11-14, Tables XI-XIV)]

*Disaggregated municipal analysis* - At the municipal level, none of the simple linear regression models demonstrated statistically significant associations between malaria cases and the independent variables eval-

TABLE II

Impact was measured in terms of public health outcomes, financial losses, logistical disruptions, or reduced effectiveness of vector control actions

| Level | Description |
|---|---|
| High | Directly compromises essential service delivery or epidemiological targets. |
| Medium | Causes delays or inefficiencies without critically affecting system function. |
| Low | Generates localized, low-cost, or easily reversible impacts. |

TABLE III

The level of risk with probability and impact
using a 3x3 matrix (high, medium, low)

| Probability ↓ | Low impact | Medium impact | High impact |
| --- | --- | --- | --- |
| Low | Low | Low | Medium |
| Medium | Low | Medium | High |
| High | Medium | High | High |

Correlation between insecticide dispensation and malaria cases in Brazil

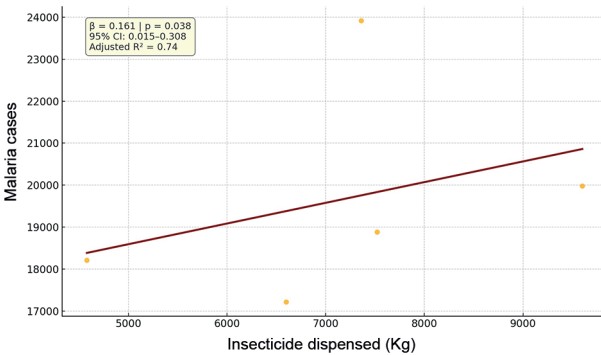

Fig. 1: relationship between insecticide dispensation and malaria cases in Brazil (2019-2023). Each yellow dot represents one year (2019-2023), with the unit of analysis being the year at the national aggregate level. The fitted line illustrates the general direction of association based on ordinary least squares (OLS), shown here for descriptive purposes only. Model assumptions (normality, autocorrelation, and homoscedasticity) were checked, and where violations were identified, robust estimators and non-parametric correlations were applied. Given the small sample size (n = 5), results should be interpreted with caution and without inference at the municipal level.

uated (insecticide dispensed, temperature, precipitation, and deforestation). In Barcelos, although deforestation and insecticide allocation showed negative coefficients, wide confidence intervals and non-significant p-values indicated weak explanatory power. In Tefé, all variables presented low coefficients and poor model fit, with adjusted $R^2$ values close to zero. Similarly, in São Gabriel da Cachoeira, insecticide allocation showed a positive but non-significant coefficient, while climatic and environmental variables also failed to reach statistical significance. Overall, the low adjusted $R^2$ values across municipalities suggest that local malaria dynamics are not adequately captured by these predictors when aggregated annually, underscoring the influence of contextual factors and management quality beyond the variables tested (Table IV).

*Specific trends* - In Barcelos, a negative coefficient was found for the relationship between insecticide use and malaria cases ($\beta = -0.94$); in Tefé, despite a significant reduction in insecticide volume in 2020 and 2021, there was a sharp increase in malaria cases, indicating logistical weaknesses; in São Gabriel da Cachoeira, a positive relationship between insecticide use and malaria cases was observed, reflecting a delayed response following increased transmission.

*Qualitative analysis of the custody chain* - The constructed risk matrix revealed structural weaknesses in the planning and application stages of the insecticide supply chain, particularly the lack of integration between epidemiological surveillance and logistics, low traceability of local chain stages, underutilization of environmental data in planning, and reduced IRS coverage in critical areas such as Indigenous communities and mining regions. Beyond the statistical evidence and predictive models employed, the risk-matrix-based qualitative evaluation provided a systemic view of operational and institutional bottlenecks, as outlined below:

(a) Planning and demand forecasting. Input planning remains largely based on historical consumption patterns and reactive municipal requests, without incorporating predictive data on epidemiological, climatic, or environmental risk. For instance, there is no systematic integration with deforestation, temperature, or prior outbreak data to anticipate IRS needs. This failure has a high impact, as it compromises timely intervention and creates scenarios of stockouts or overstocking. The probability of occurrence is high and has been observed in several operational cycles (classified as high risk).

(b) Storage and transport. Inadequate storage conditions were found in the evaluated municipalities, including lack of temperature control, absence of chemical segregation, and poor infrastructure. Transport was often decentralized and improvised, relying on local logistics and sporadic availability of land or river transport, leading to critical delays, particularly in hard-to-reach areas.

Identified risk: compromised product integrity and delayed control actions; classification: medium probability and high impact (high risk).

(c) Application and coverage. IRS implementation faces limitations in geographic coverage and frequency. Many critical areas, such as Indigenous communities and mining zones, are not systematically covered. Additionally, the absence of post-application efficacy assessments hinders tactical adjustments and institutional learning.

Identified risk: low effectiveness of vector control actions and persistence of endemic foci; classification: high probability and high impact (high risk).

(d) Monitoring and evaluation. Monitoring of the custody chain stages is incipient and fragmented. There is no interoperability between systems such as Sivep-Malaria, SIES, and state platforms like FVS-AM, hampering integrated analysis of applied inputs and epidemiological outcomes. Moreover, systematic monitoring of vector resistance to the insecticide is nonexistent, which may undermine IRS effectiveness in the medium term.

Identified risk: lack of operational feedback and strategy adaptation to local changes; classification: high probability and high impact (high risk).

(e) Governance and federal coordination. Coordination among federal, state, and municipal levels is limited, with poorly defined responsibilities, non-transparent transfers, and scarce accountability for outcomes. The absence of collegiate decision-making bodies focused on logistical chain management exacerbates coordination issues.

TABLE IV

Results of the simple linear regression models for the municipalities of Barcelos, Tefé,
and São Gabriel da Cachoeira, examining four independent variables

| Municipality | Variable | Coefficient (β) | p-value | 95% CI (Lower) | 95% CI (Upper) | Adjusted R² |
|---|---|---|---|---|---|---|
| Barcelos | Insecticide (kg) | -0,94 | 0,21 | -2,72 | 0,84 | 0,34 |
| | Temperature (°C) | 542,1 | 0,44 | -1110 | 2194 | 0,27 |
| | Precipitation (mm) | -30,2 | 0,32 | -102,3 | 41,9 | 0,35 |
| | Deforestation (km²) | -168,4 | 0,22 | -479,6 | 142,7 | 0,43 |
| Tefé | Insecticide (kg) | -0,15 | 0,67 | -1,18 | 0,88 | 0,04 |
| | Temperature (°C) | 634,5 | 0,29 | -702,2 | 1971,2 | 0,36 |
| | Precipitation (mm) | -11,7 | 0,57 | -59,2 | 35,9 | 0,12 |
| | Deforestation (km²) | 78,3 | 0,42 | -153,4 | 310,0 | 0,29 |
| São Gabriel da Cachoeira | Insecticide (kg) | 0,66 | 0,28 | -0,76 | 2,08 | 0,27 |
| | Temperature (°C) | -542,0 | 0,51 | -2021 | 910,1 | 0,23 |
| | Precipitation (mm) | 26,1 | 0,40 | -52,3 | 104,6 | 0,31 |
| | Deforestation (km²) | 322,0 | 0,37 | -546,3 | 1190 | 0,33 |

CI: confidence interval.

Identified risk: uncoordinated decisions, inefficient responses, and territorial inequality in vector control; classification: high probability and high impact (high risk).

*Conceptual model of the insecticide chain of custody* - Fig. 2 shows the conceptual model developed to represent the main operational and relational stages of the Etofenproxy 20% PM insecticide chain of custody, as used in the IRS malaria control strategy. The model is structured as a logical flow covering everything from centralised input acquisition to final field application, including the critical phases of planning, resource allocation, storage, distribution and vector control action execution. The model also incorporates a feedback component between the malaria case load and input scheduling, illustrating the difference between a reactive approach (based on an increase in cases) and a predictive approach (based on environmental and epidemiological risk factors).

Additionally, the flowchart identifies critical decision points and potential operational bottlenecks that could undermine the effectiveness of vector control, including logistical delays, inadequate entomological monitoring, federal coordination failures, and a disconnect between surveillance and logistics systems. The aim of this representation is to provide a systemic view of how the chain works, and to support the proposal to restructure it based on operational intelligence, data interoperability, and territorialised governance.

## DISCUSSION

The findings of this study suggest that the custody chain of Etofenprox 20% PM, used in the IRS strategy, presents important weaknesses in planning, federal coordination, and integration with epidemiological surveillance. At the national level, a statistically significant association was observed between the volume of insecti-

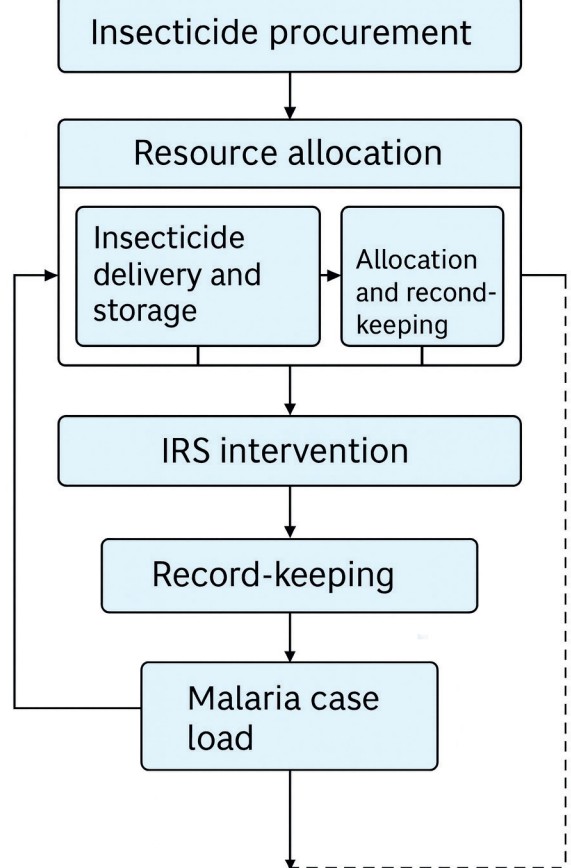

Fig. 2: conceptual flowchart of the insecticide custody chain in malaria vector control, representing the current process ("as is") from forecasting to application and record-keeping. This diagram is descriptive of operational steps only and does not illustrate effects or impacts. Source: Prepared by the author based on literature and study findings.

cide dispensed and malaria cases (β = 0.161; p = 0.038); however, this relationship was not consistent in municipal-level analyses, indicating that the effectiveness of allocation is strongly influenced by local context and management quality. Despite the constraints of limited sample size and data granularity, the results highlight the potential of time-series approaches as complementary tools for predictive planning of vector control supplies, contributing to more anticipatory and adaptive responses in areas of high environmental receptivity.

The absence of consistent statistical associations at the municipal level should be interpreted with caution, as the unit of analysis was the year, limiting the ability to capture short-term dynamics. In São Gabriel da Cachoeira, for instance, the temporal overlap between increases in insecticide distribution and rises in malaria cases suggests that allocation may have occurred in response to, rather than in anticipation of, epidemiological changes. In Barcelos and Tefé, the irregular patterns between insecticide use and annual case incidence indicate potential misalignment between intervention timing and local epidemiological trends. These observations, while exploratory, highlight the need for more granular data to assess how operational decisions align with local transmission dynamics.

These patterns contradict international guidelines emphasizing the importance of an integrated, predictive, and risk-adapted approach for malaria vector control.[3] The WHO recommends using climatic and environmental data as inputs for strategic planning, particularly in high-receptivity areas such as dense forests, informal settlements, and Indigenous territories.[3,5]

The statistical analysis revealed a positive correlation between deforestation and malaria cases, with national-level significance (β = 0.626; p = 0.034), and similar trends in Barcelos and São Gabriel da Cachoeira. These findings are consistent with studies linking increased deforestation to greater human exposure to vectors in ecological transition zones and areas of anthropogenic expansion.[10,11] Therefore, incorporating deforestation data into predictive models may serve as a strategic tool to improve vector control planning.

The developed risk matrix identified critical vulnerabilities in the areas of planning, storage, and application. The lack of interoperability among systems such as Sivep-Malaria, SIES, and state logistics platforms impedes proper tracking of insecticide distribution and coverage. Furthermore, the absence of monitoring for vector resistance to the insecticide prevents medium-term evaluations of effectiveness, as recommended by the PNEM.[4,12-14]

Another critical point is the fragility of federal coordination. IRS logistics depend on centralized transfers and decentralized requests, with no clear accountability mechanisms or integration among governance levels. This structural gap compromises territorial equity and perpetuates operational inequalities between municipalities with varying technical and administrative capacities.

It is also important to note that this study has several limitations that must be considered when interpreting the results. First, only secondary data from official systems (Sivep-Malaria, SIES, FVS-AM, INMET) were used. Although institutionally validated, these sources are subject to underreporting, delays, and inconsistencies, particularly in the insecticide logistics records. Additionally, the five-year analysis period (2019-2023) may not fully capture broader epidemiological cycles or long-term environmental variability. The absence of entomological efficacy data, such as vector resistance to pyrethroids and effective IRS coverage indices, also limits the ability to infer the direct impact of interventions on transmission. Moreover, gaps or discontinuities in logistics records for some municipalities and years compromised the completeness of the analysis.

Despite these limitations, the findings provide relevant insights for improving IRS logistical governance. It is recommended that the Ministry of Health establish formal intergovernmental agreements with shared goals, performance indicators, and conditional funding linked to system integration and transparency in insecticide use data. The implementation of federative control panels interoperable with epidemiological and environmental surveillance systems could enable anticipatory logistical planning based on climatic and environmental risk. Finally, strengthening entomological surveillance and logistical chain traceability, including regular audits and local team training, is essential to ensure rational use of supplies and progress toward the goals of the PNEM.

*Policy implications* - Based on the findings, there is an urgent need to restructure the insecticide custody chain around four strategic pillars:

(1) Predictive and integrated planning: incorporate models that combine epidemiological, environmental (climate and deforestation), and social data to guide supply demand.

(2) System interoperability: integrate platforms such as Sivep-Malaria, SIES, and state logistics registries to enhance traceability and operational intelligence.

(3) Strengthened entomological surveillance: ensure regular testing of vector susceptibility to insecticides and adjust strategies according to resistance patterns.

(4) Decentralization with shared governance: establish intergovernmental agreements with targets, indicators, and co-financing mechanisms to ensure equitable coverage in priority areas.

In conclusion, this study highlights structural weaknesses in the custody chain of Etofenprox 20% PM, used in malaria vector control in the Brazilian Amazon from 2019 to 2023. Quantitative analysis revealed significant national-level correlations, particularly between insecticide volume and malaria incidence, as well as between deforestation and increased cases. However, the lack of statistically significant associations at the municipal level indicates a reactive management pattern with limited responsiveness to local epidemiological and environmental dynamics.

Failures in information system integration, lack of effective entomological surveillance, and limited federal coordination undermine the effectiveness of IRS. In this context, it is recommended that the custody chain be restructured based on evidence, incorporating predictive

models, data interoperability, planned decentralization, and continuous monitoring of insecticide effectiveness.

The most notable quantitative findings show that, at the national level, insecticide volume is significantly correlated with malaria incidence, and that deforestation emerges as one of the main environmental predictors of transmission, with high explanatory capacity. However, at the municipal level, operational fragmentation and lack of coordinated planning dilute the effectiveness of these relationships, indicating shortcomings in local strategy adaptation. This scenario underscores the urgency of reformulating vector control policy based on integrated evidence.

Public policies should systematically incorporate environmental, climatic, and epidemiological data to enable outbreak forecasting and efficient resource allocation. Only with a risk-oriented, science-based approach will it be possible to sustainably advance toward malaria elimination in Brazil.

To strengthen the custody chain, measures should include: (i) planning and forecasting with time-series models, environmental data, and intergovernmental agreements; (ii) storage and transport improvements through infrastructure, training, and monitoring; (iii) application and coverage expansion with workforce support and mobile reporting; (iv) monitoring and evaluation using efficacy indicators, resistance testing, and interoperable systems; and (v) governance and coordination via permanent committees, shared management agreements, and contingency plans. Overall, malaria elimination in Brazil depends not only on reliable supplies, but also on integrated governance capable of anticipating risks, responding rapidly, and ensuring equitable and universal coverage.

## AUTHORS' CONTRIBUTION

In developing this article, MD served as the lead author, being responsible for the study's conception, methodological design, data analysis and interpretation, as well as drafting and critically revising the manuscript; MC and JL, as supervisors, contributed with technical and scientific guidance, intellectual review, and validation of the analyses presented. The co-authors RS, CP, CK and DC provided technical support, assisting in the operational stages of data collection, organization, and contributing to the final review of the manuscript. This manuscript does not contain any individual person's data in any form (including individual details, images, or videos).

## DATA AVAILABILITY

All datasets analyzed during the current study are publicly available through official Brazilian government information systems. Malaria case data were obtained from the Malaria Epidemiological Surveillance Information System (SIVEP-Malaria). Insecticide dispensation data were accessed through the Strategic Health Supplies Information System (SIES/MS) and the Amazonas State Health Surveillance Foundation (FVS-AM). Demographic data were derived from the Brazilian Institute of Geography and Statistics (IBGE), while meteorological data (temperature and precipitation) were sourced from the National Institute of Meteorology (INMET). Deforestation data were retrieved from the TerraBrasilis platform of the National Institute for Space Research (INPE).

All datasets are open access and available from:

SIVEP-Malaria: https://public.tableau.com/app/profile/mal.ria.brasil

SIES/MS:https://www.gov.br/pt-br/servicos/solicitar-acesso-a-informacao-no-servico-de-informacao-ao-cidadao-do-ministerio-da-saude-sic-ms

IBGE: https://www.ibge.gov.br

INMET: https://bdmep.inmet.gov.br/

TerraBrasilis/INPE: https://terrabrasilis.dpi.inpe.br

All data used are aggregated and de-identified, with no personal information included. Analytical scripts and supplementary materials generated during this study are available from the corresponding author upon reasonable request.

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

# OPEN PEER REVIEW

Memórias do IOC thanks the anonymous reviewers for their contribution to the peer review of this work.

**FIRST REVIEW ROUND**

<div align="right">REVIEWERS' COMMENTS</div>

**REVIEWER #1**

Thank you for the opportunity to review this manuscript, which presents an important topic related to malaria surveillance and the assessment of the insecticide chain of custody in some municipalities in the Amazon. The manuscript addresses a relevant issue. However, several aspects require clarification and revision to improve the overall clarity, coherence, and scientific rigor of the work. Please find below my detailed comments:

The use of ARIMA models in the manuscript requires further clarification. ARIMA is primarily designed for analyzing time series data to evaluate trends and make forecasts, rather than for assessing associations between variables. Please revise the manuscript to clearly articulate the rationale for using ARIMA in the context of your analysis, or consider alternative statistical approaches better suited for evaluating associations.

The criteria for selecting municipalities are not fully justified. To ensure minimal statistical assumptions, the selection process should ideally involve randomization or pairing of municipalities to enable more robust comparisons, including potential controls. The phrase "epidemiological relevance" is vague; please specify the parameters used for selection.

In line 180 and elsewhere, the incidence rate should be used instead of the absolute number of cases, which would provide a more accurate and comparable epidemiological assessment.

The current description of the "chain of custody" appears to extend beyond its conventional definition. Specifically, attributing outcomes such as mosquito mortality or effectiveness to the chain may be misleading. Please clarify or adjust the scope of the flowchart (Figure 1) and reconsider the title and descriptors used.

The manuscript includes several statements that imply causality or broader extrapolation from a limited dataset. Given the small number of municipalities and the unit of analysis being years rather than locations, such inferences should be carefully moderated. This includes conclusions drawn in Figure 2, and statements made in line 378.

Minor Comments and Editorial Suggestions

Please ensure that figure numbering follows a sequential and logical order. There seems to be a mismatch regarding figure numbering — please verify and correct as needed.

Consider replacing or joining redundant or repetitive sentences. Some of them are marked in the text as comments to the PDF.

The sentence in lines 101–107 would fit better in the concluding part of the introduction. Consider merging it with the paragraph spanning lines 119–125.

The paragraph 115–118 contain methodological content and should be moved to the Methods section.

Paragraphs 119–125 and 126–131 both touch on study objectives or methods and should be integrated with others as suggested in the PDF annotations.

The conclusion should be more concise and written in a paragraph form rather than using bullet points. Please ensure it focuses on the key findings without overextending the implications beyond the data.

Figure 2 (Correlation Plot)

Given the small sample size, the title of the figure should avoid suggesting broad extrapolations.

Major Revision

The manuscript presents significant value, particularly in its case reporting and practical implications for malaria control. However, key methodological clarifications, structural improvements, and cautious interpretation of findings are needed before the manuscript can be considered for publication.

<div align="right">AUTHORS' RESPONSE TO THE REVIEWERS</div>

Dear Editor and Reviewers,

We thank you for the careful reading of our manuscript and the constructive comments provided. We have revised the text thoroughly, following each suggestion. Below, we provide a point-by-point response, indicating how we addressed every comment. All changes are highlighted in the revised manuscript.

Reviewer's Comments and Authors' Responses

1. "Reported cases are especially relevant to the epidemiology analysis."

Response: Abstract - We thank the reviewer for this important observation. We would like to clarify that our intention was not to discredit the value of reported cases for epidemiological analysis. Instead, we aimed to emphasise that, for an efficient custody chain, it is necessary to consider multiple variables that are part of the strategic management process of malaria control insecticides. While reported cases are crucial for epidemiological

surveillance, the custody chain evaluation requires integrating additional operational and logistical factors (such as forecasting, procurement, storage, distribution, and application), in order to strengthen the efficiency and responsiveness of malaria control programs.

2. "ARIMA models are not intended to assess associations… Please, clarify the use of ARIMA models."

Response: Abstract - ARIMA was applied exclusively to explore temporal patterns (trend, stationarity, lags). Associations between variables were evaluated with regression models.

3. "Are the figures expected to appear in a sequential ordering? Should it be the supp fig 2?"

Response: All article - We corrected the numbering and ensured sequential order of Supplementary Figures.

4. "This sentence is somehow already stated in lines 74–77. Please put ideas together."

Response: Introduction - Redundant sentences were merged into a single, concise statement in the Introduction.

5. "None of these include deforestation as a variable."

Response: Introduction - We revised the text to clarify that deforestation is drawn from the scientific literature and proposed as a complementary predictive variable by the authors, not included in national or WHO operational guidelines.

6. "This sentence should be in the last paragraph (119–125)."

Response: Introduction - Sentence repositioned to the indicated paragraph for logical flow.

7–10. "This paragraph sounds like a method. Move it to Methods and rephrase." / "This paragraph should also be moved…" / "I suggest the authors merge this paragraph with the previous one…" / "Methods. Again, I suggest moving it…"

Response: Introduction - All methodological content was moved to Materials and Methods and rephrased accordingly. Objectives and hypotheses were consolidated in the Introduction.

11. "There's a clear definition of effectiveness… avoid direct conclusions since effectiveness was not assessed here."

Response: Materials and Methods - We adjusted wording: replaced "effectiveness" with "operational efficiency/ potential reach" and consistently used "exploratory" when referring to impacts.

12. "This paragraph is redundant… consequence of the first two hypotheses."

Response: Materials and Methods - We deleted the redundant paragraph and integrated its content with the hypotheses.

13. (Marked as "strikethrough")

Response: Materials and Methods - The stricken text was removed.

14. "The incidence should be used instead."

Response: Indicators and Variables - In the Brazilian context, the absolute number of malaria cases is a particularly relevant indicator for epidemiological analysis and programmatic decision-making. This is because the disease burden is highly concentrated in the Amazon region, where small municipalities with reduced populations can still present hundreds or thousands of cases annually. In such scenarios, incidence rates (API) may overestimate the risk in low-population settings or underestimate the operational demand in larger ones, while the absolute number of cases more accurately reflects the real workload for health services. Moreover, national surveillance systems such as SIVEP-Malária are structured around case counts, which are the basis for logistics, allocation of strategic supplies, and outbreak response. Therefore, although API is essential for comparative analyses between different territories, the absolute number of cases remains the most practical and informative measure to guide malaria control in Brazil.

15. "Already stated before. I suggest deleting."

Response: Statistical Analysis - Duplicate definitions were deleted, keeping only the necessary description of the risk matrix.

16. "flowchart"

Response: Conceptual model of the insecticide chain of custody - We renamed the figure as "Conceptual flowchart of the custody chain" and updated its caption.

17–19. "The flowchart only shows the current chain…" / "I'm not sure the effect on mosquitoes is part…" / "Is it part of the results?"

Response: Conceptual model of the insecticide chain of custody - We clarified that the flowchart represents the current custody chain (as is). References to entomological effects were removed. The figure was relocated to Methods/Supplementary Material as a conceptual tool.

20. "Malaria incidence or number of cases? Also, is there a need for this analysis…?"

Response: Results - In the Brazilian context, the absolute number of malaria cases is a particularly relevant indicator for epidemiological analysis and programmatic decision-making. This is because the disease burden is highly concentrated in the Amazon region, where small municipalities with reduced populations can still present hundreds or thousands of cases annually. In such scenarios, incidence rates (API) may overestimate the risk in low-population settings or underestimate the operational demand in larger ones, while the absolute number of cases more accurately reflects the real workload for health services. Moreover, national surveillance systems such as SIVEP-Malária are structured around case counts, which are the basis for logistics, allocation of strategic supplies, and outbreak response. Therefore, although API is essential for comparative analyses between different territories, the absolute number of cases remains the most practical and informative measure to guide malaria control in Brazil.

We thank the reviewer for this important question. We agree that the primary focus of the study is the custody chain of insecticides. However, malaria case data remain central to this assessment because they provide the epidemiological context that justifies and guides the custody chain. Without considering the burden of disease, it would not be possible to evaluate whether the allocation, storage, and use of strategic inputs are aligned with actual needs. Case analysis allows us to identify whether the custody chain operates proactively or reactively, and whether it effectively anticipates or merely responds to transmission peaks. Therefore, while our study does not aim to establish causal links, the inclusion of malaria case data is essential to assess the performance and adequacy of the custody chain as a strategic component of malaria control.

21. "Considering the very limited number of municipalities, avoid extrapolations."

Response: Results - Language was softened; interpretations were limited strictly to the three study municipalities. No national generalizations were retained.

22. "Please, clarify what the yellow points mean… assumptions violated."

Response: Results - Figure legend was clarified (yellow points = years 2019–2023). We corrected text to avoid implying inter-municipality comparisons. Model assumptions were checked; where violated, we used robust/non-parametric approaches or descriptive reporting without p-values.

23. "The figures showing the time-series should be mentioned here."

Response: Results - Supplementary time-series figures are now explicitly cited in the Results section.

24. "This entire sentence sounds like discussion and should be moved."

Response: Results - Sentence moved to Discussion.

25. "As a suggestion, explore time-series cross-correlation."

Response: Results - We followed the suggestion and added a cross-correlation function (CCF) analysis between incidence and environmental/operational variables, presented in Supplementary Material with brief mention in Methods and Results.

26. "Figure 2?"

Response: All figure numbering (main and supplementary) was revised and corrected.

27. "Authors should begin by going directly to the description of the results… a paragraph may be missing here."

Response: Disaggregated Municipal Analysis - A new summary paragraph was added before Table 2, highlighting key trends and non-significant results.

28. "As stated before, this cannot be concluded… unit of analysis is year."

Response: Discussion - The inappropriate conclusion was removed; interpretations are consistent with the year as the unit of analysis.

29. "This section should be shortened and sound more direct. Try to avoid bullet points."

Response: Conclusion - The Conclusion/Implications section was rewritten in concise prose, without bullet points.

Additional global changes

• Consistent use of absolute number of malaria cases as the main indicator, as justified in the manuscript. This choice reflects the operational reality of malaria control in Brazil, where case counts are the basis for surveillance, logistics, and custody chain management, while incidence (API) is used in complementary analyses.

• Clear distinction between time-series description (ARIMA) and associations (regression).

• Streamlined Introduction; all Methods content centralized in Materials and Methods.

• Figures renumbered; legends clarified; redundant text removed.

• Language moderated to avoid over-extrapolation.

• Discussion more focused; Conclusions shortened and made more direct.

Final Statement

We believe these revisions have significantly improved the manuscript, addressing all concerns raised. We thank the reviewers for their insightful feedback, which has strengthened the scientific rigor and clarity of our work.

Sincerely,

## SECOND REVIEW ROUND

### REVIEWERS' COMMENTS

### REVIEWER #1

The authors made the necessary changes to the manuscript in accordance with the reviewers' recommendations. However, I think some very minor issues must be solved before the manuscript can be considered for publication. Please review all the figures and their captions for any remaining Portuguese wording.

For instance, some figures mention "Casos Nacionais de Malária ", "Casos de malária do estado do Amazonas", "Compra de inseticidas e casos de Malária", "Temperatura e precipitação", the entire Figure 4 legend, Table 10 - "variação", Table 12 - "malária", and so on...

