## [Reviewer Report · FIRST REVIEW ROUND - REVIEWERS COMMENTS]

## REVIEWER #1

Thank you for the opportunity to review this manuscript, which presents an important topic related to malaria surveillance and the assessment of the insecticide chain of custody in some municipalities in the Amazon. The manuscript addresses a relevant issue. However, several aspects require clarification and revision to improve the overall clarity, coherence, and scientific rigor of the work. Please find below my detailed comments:

The use of ARIMA models in the manuscript requires further clarification. ARIMA is primarily designed for analyzing time series data to evaluate trends and make forecasts, rather than for assessing associations between variables. Please revise the manuscript to clearly articulate the rationale for using ARIMA in the context of your analysis, or consider alternative statistical approaches better suited for evaluating associations.

The criteria for selecting municipalities are not fully justified. To ensure minimal statistical assumptions, the selection process should ideally involve randomization or pairing of municipalities to enable more robust comparisons, including potential controls. The phrase “epidemiological relevance” is vague; please specify the parameters used for selection.

In line 180 and elsewhere, the incidence rate should be used instead of the absolute number of cases, which would provide a more accurate and comparable epidemiological assessment.

The current description of the “chain of custody” appears to extend beyond its conventional definition. Specifically, attributing outcomes such as mosquito mortality or effectiveness to the chain may be misleading. Please clarify or adjust the scope of the flowchart (Figure 1) and reconsider the title and descriptors used.

The manuscript includes several statements that imply causality or broader extrapolation from a limited dataset. Given the small number of municipalities and the unit of analysis being years rather than locations, such inferences should be carefully moderated. This includes conclusions drawn in Figure 2, and statements made in line 378.

**Minor Comments and Editorial Suggestions**

Please ensure that figure numbering follows a sequential and logical order. There seems to be a mismatch regarding figure numbering — please verify and correct as needed.

Consider replacing or joining redundant or repetitive sentences. Some of them are marked in the text as comments to the PDF.

The sentence in lines 101–107 would fit better in the concluding part of the introduction. Consider merging it with the paragraph spanning lines 119–125.

The paragraph 115–118 contain methodological content and should be moved to the Methods section.

Paragraphs 119–125 and 126–131 both touch on study objectives or methods and should be integrated with others as suggested in the PDF annotations.

The conclusion should be more concise and written in a paragraph form rather than using bullet points. Please ensure it focuses on the key findings without overextending the implications beyond the data.

Figure 2 (Correlation Plot): Given the small sample size, the title of the figure should avoid suggesting broad extrapolations.

**Major Revision**

The manuscript presents significant value, particularly in its case reporting and practical implications for malaria control. However, key methodological clarifications, structural improvements, and cautious interpretation of findings are needed before the manuscript can be considered for publication.

## AUTHORS' RESPONSE TO THE REVIEWERS

Dear Editor and Reviewers,

We thank you for the careful reading of our manuscript and the constructive comments provided. We have revised the text thoroughly, following each suggestion. Below, we provide a point-by-point response, indicating how we addressed every comment. All changes are highlighted in the revised manuscript.

1. “Reported cases are especially relevant to the epidemiology analysis.”

Response: Abstract - We thank the reviewer for this important observation. We would like to clarify that our intention was not to discredit the value of reported cases for epidemiological analysis. Instead, we aimed to emphasise that, for an efficient custody chain, it is necessary to consider multiple variables that are part of the strategic management process of malaria control insecticides. While reported cases are crucial for epidemiological surveillance, the custody chain evaluation requires integrating additional operational and logistical factors (such as forecasting, procurement, storage, distribution, and application), in order to strengthen the efficiency and responsiveness of malaria control programs.

2. “ARIMA models are not intended to assess associations… Please, clarify the use of ARIMA models.”

Response: Abstract - ARIMA was applied exclusively to explore temporal patterns (trend, stationarity, lags). Associations between variables were evaluated with regression models.

3. “Are the figures expected to appear in a sequential ordering? Should it be the supp fig 2?”

Response: All article - We corrected the numbering and ensured sequential order of Supplementary Figures.

4. “This sentence is somehow already stated in lines 74–77. Please put ideas together.”

Response: Introduction - Redundant sentences were merged into a single, concise statement in the Introduction.

5. “None of these include deforestation as a variable.”

Response: Introduction - We revised the text to clarify that deforestation is drawn from the scientific literature and proposed as a complementary predictive variable by the authors, not included in national or WHO operational guidelines.

6. “This sentence should be in the last paragraph (119–125).”

Response: Introduction - Sentence repositioned to the indicated paragraph for logical flow.

7–10. “This paragraph sounds like a method. Move it to Methods and rephrase.” / “This paragraph should also be moved…” / “I suggest the authors merge this paragraph with the previous one…” / “Methods. Again, I suggest moving it…”

Response: Introduction - All methodological content was moved to Materials and Methods and rephrased accordingly. Objectives and hypotheses were consolidated in the Introduction.

11. “There’s a clear definition of effectiveness… avoid direct conclusions since effectiveness was not assessed here.”

Response: Materials and Methods - We adjusted wording: replaced “effectiveness” with “operational efficiency/potential reach” and consistently used “exploratory” when referring to impacts.

12. “This paragraph is redundant… consequence of the first two hypotheses.”

Response: Materials and Methods - We deleted the redundant paragraph and integrated its content with the hypotheses.

13. (Marked as “strikethrough”)

Response: Materials and Methods - The stricken text was removed.

14. “The incidence should be used instead.”

Response: Indicators and Variables - In the Brazilian context, the absolute number of malaria cases is a particularly relevant indicator for epidemiological analysis and programmatic decision-making. This is because the disease burden is highly concentrated in the Amazon region, where small municipalities with reduced populations can still present hundreds or thousands of cases annually. In such scenarios, incidence rates (API) may overestimate the risk in low-population settings or underestimate the operational demand in larger ones, while the absolute number of cases more accurately reflects the real workload for health services. Moreover, national surveillance systems such as SIVEP-Malária are structured around case counts, which are the basis for logistics, allocation of strategic supplies, and outbreak response. Therefore, although API is essential for comparative analyses between different territories, the absolute number of cases remains the most practical and informative measure to guide malaria control in Brazil.

15. “Already stated before. I suggest deleting.”

Response: Statistical Analysis - Duplicate definitions were deleted, keeping only the necessary description of the risk matrix.

16. “flowchart”

Response: Conceptual model of the insecticide chain of custody - We renamed the figure as “Conceptual flowchart of the custody chain” and updated its caption.

17–19. “The flowchart only shows the current chain…” / “I’m not sure the effect on mosquitoes is part…” / “Is it part of the results?”

Response: Conceptual model of the insecticide chain of custody - We clarified that the flowchart represents the current custody chain (as is). References to entomological effects were removed. The figure was relocated to Methods/Supplementary Material as a conceptual tool.

20. “Malaria incidence or number of cases? Also, is there a need for this analysis…?”

Response: Results - In the Brazilian context, the absolute number of malaria cases is a particularly relevant indicator for epidemiological analysis and programmatic decision-making. This is because the disease burden is highly concentrated in the Amazon region, where small municipalities with reduced populations can still present hundreds or thousands of cases annually. In such scenarios, incidence rates (API) may overestimate the risk in low-population settings or underestimate the operational demand in larger ones, while the absolute number of cases more accurately reflects the real workload for health services. Moreover, national surveillance systems such as SIVEP-Malária are structured around case counts, which are the basis for logistics, allocation of strategic supplies, and outbreak response. Therefore, although API is essential for comparative analyses between different territories, the absolute number of cases remains the most practical and informative measure to guide malaria control in Brazil.

We thank the reviewer for this important question. We agree that the primary focus of the study is the custody chain of insecticides. However, malaria case data remain central to this assessment because they provide the epidemiological context that justifies and guides the custody chain. Without considering the burden of disease, it would not be possible to evaluate whether the allocation, storage, and use of strategic inputs are aligned with actual needs. Case analysis allows us to identify whether the custody chain operates proactively or reactively, and whether it effectively anticipates or merely responds to transmission peaks. Therefore, while our study does not aim to establish causal links, the inclusion of malaria case data is essential to assess the performance and adequacy of the custody chain as a strategic component of malaria control.

21. “Considering the very limited number of municipalities, avoid extrapolations.”

Response: Results - Language was softened; interpretations were limited strictly to the three study municipalities. No national generalizations were retained.

22. “Please, clarify what the yellow points mean… assumptions violated.”

Response: Results - Figure legend was clarified (yellow points = years 2019–2023). We corrected text to avoid implying inter-municipality comparisons. Model assumptions were checked; where violated, we used robust/non-parametric approaches or descriptive reporting without p-values.

23. “The figures showing the time-series should be mentioned here.”

Response: Results - Supplementary time-series figures are now explicitly cited in the Results section.

24. “This entire sentence sounds like discussion and should be moved.”

Response: Results - Sentence moved to Discussion.

25. “As a suggestion, explore time-series cross-correlation.”

Response: Results - We followed the suggestion and added a cross-correlation function (CCF) analysis between incidence and environmental/operational variables, presented in Supplementary Material with brief mention in Methods and Results.

26. “Figure 2?”

Response: All figure numbering (main and supplementary) was revised and corrected.

27. “Authors should begin by going directly to the description of the results… a paragraph may be missing here.”

Response: Disaggregated Municipal Analysis - A new summary paragraph was added before Table 2, highlighting key trends and non-significant results.

28. “As stated before, this cannot be concluded… unit of analysis is year.”

Response: Discussion - The inappropriate conclusion was removed; interpretations are consistent with the year as the unit of analysis.

29. “This section should be shortened and sound more direct. Try to avoid bullet points.”

Response: Conclusion - The Conclusion/Implications section was rewritten in concise prose, without bullet points.

**Additional global changes**

• Consistent use of absolute number of malaria cases as the main indicator, as justified in the manuscript. This choice reflects the operational reality of malaria control in Brazil, where case counts are the basis for surveillance, logistics, and custody chain management, while incidence (API) is used in complementary analyses.

• Clear distinction between time-series description (ARIMA) and associations (regression).

• Streamlined Introduction; all Methods content centralized in Materials and Methods.

• Figures renumbered; legends clarified; redundant text removed.

• Language moderated to avoid over-extrapolation.

• Discussion more focused; Conclusions shortened and made more direct.

**Final Statement**

We believe these revisions have significantly improved the manuscript, addressing all concerns raised. We thank the reviewers for their insightful feedback, which has strengthened the scientific rigor and clarity of our work.

Sincerely,

---

## [Reviewer Report · SECOND REVIEW ROUND - REVIEWERS COMMENTS]

## REVIEWER #1

The authors made the necessary changes to the manuscript in accordance with the reviewers’ recommendations.

However, I think some very minor issues must be solved before the manuscript can be considered for publication.

Please review all the figures and their captions for any remaining Portuguese wording.

For instance, some figures mention “Casos Nacionais de Malária “, “Casos de malária do estado do Amazonas”, “Compra de inseticidas e casos de Malária”, “Temperatura e precipitação”, the entire Figure 4 legend, Table 10 - “variação”, Table 12 - “malária”, and so on...